# ON NOISE INJECTION IN GENERATIVE ADVERSARIAL NETWORKS

## ABSTRACT

Noise injection is an effective way of circumventing overfitting and enhancing generalization in machine learning, the rationale of which has been validated in deep learning as well. Recently, noise injection exhibits surprising effectiveness when generating high-fidelity images in Generative Adversarial Networks (e.g. StyleGAN). Despite its successful applications in GANs, the mechanism of its validity is still unclear. In this paper, we propose a geometric framework to theoretically analyze the role of noise injection in GANs. Based on Riemannian geometry, we successfully model the noise injection framework as fuzzy equivalence on geodesic normal coordinates. Guided by our theories, we find that existing methods are incomplete and a new strategy for noise injection is devised. Experiments on image generation and GAN inversion demonstrate the superiority of our method.

## 1 INTRODUCTION

Noise injection is usually applied as regularization to cope with overfitting or facilitate generalization in neural networks (Bishop, 1995; An, 1996). The effectiveness of this simple technique has also been proved in various tasks in deep learning, such as learning deep architectures (Hinton et al., 2012; Srivastava et al., 2014; Noh et al., 2017), defending adversarial attacks (He et al., 2019), facilitating stability of differentiable architecture search with reinforcement learning (Liu et al., 2019; Chu et al., 2020), and quantizing neural networks (Baskin et al., 2018). In recent years, noise injection[1] has attracted more and more attention in the community of Generative Adversarial Networks (GANs) (Goodfellow et al., 2014a). Extensive research shows that it helps stabilize the training procedure (Arjovsky & Bottou, 2017; Jenni & Favaro, 2019) and generate images of high fidelity (Karras et al., 2019a;b; Brock et al., 2018). In practice, Fig. 1 shows significant improvement in hair quality due to noise injection.

Particularly, noise injection in StyleGAN (Karras et al., 2019a;b) has shown the amazing capability of helping generate sharp details in images, shedding new light on obtaining high-quality photo-realistic results using GANs. Therefore, studying the underlying principle of noise injection in GANs is an important theoretical work of understanding GAN algorithms. In this paper, we propose a theoretical framework to explain and improve the effectiveness of noise injection in GANs. Our framework is motivated from a geometric perspective and also combined with the results of optimal transportation problem in GANs (Lei et al., 2019a;b). Our contributions are listed as follows:

- We show that the existing GAN architectures, including Wasserstein GANs (Arjovsky et al., 2017), may suffer from adversarial dimension trap, which severely penalizes the property of generator;

- Based on our theory, we attempt to explain the properties that noise injection is applied in the related literatures;

- Based on our theory, we propose a more proper form for noise injection in GANs, which can overcome the adversarial dimension trap. Experiments on the state-of-the-art GAN architecture, StyleGAN2 (Karras et al., 2019b), demonstrate the superiority of our new method compared with original noise injection used in StyleGAN2.

---

[1]It suffices to note that noise injection here is totally different from the research field of adversarial attacks raised in Goodfellow et al. (2014b).

$\longrightarrow$ Increasing noise injection depth      Standard Dev

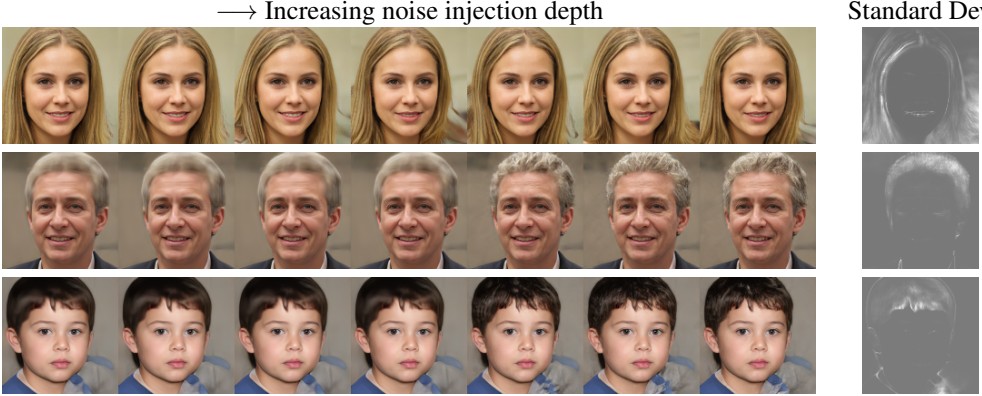

Figure 1: Noise injection significantly improves the detail quality of generated images. From left to right, we inject extra noise to the generator layer by layer. We can see that hair quality is clearly improved. Varying the injected noise and visualizing the standard deviation over 100 different seeds, we can find that the detail information such as hair, parts of background, and silhouettes are most involved, while the global information such as identity and pose is less affected.

To the best of our knowledge, this is the first work that theoretically draws the geometric picture of noise injection in GANs.

## 2 RELATED WORKS

The main drawbacks of GANs are unstable training and mode collapse. Arjovsky et al. (Arjovsky & Bottou, 2017) theoretically analyze that noise injection directly to the image space can help smooth the distribution so as to stabilize the training procedure. The authors of Distribution-Filtering GAN (DFGAN) (Jenni & Favaro, 2019) then put this idea into practice and prove that this technique will not influence the global optimality of the real data distribution. However, as the authors pointed out in (Arjovsky & Bottou, 2017), this method depends on the amount of noise. Actually, our method of noise injection is essentially different from these ones. Besides, they do not provide a theoretical vision of explaining the interactions between injected noises and features.

BigGAN (Brock et al., 2018) splits input latent vectors into one chunk per layer and projects each chunk to the gains and biases of batch normalization in each layer. They claim that this design allows direct influence on features at different resolutions and levels of hierarchy. StyleGAN (Karras et al., 2019a) and StyleGAN2 (Karras et al., 2019b) adopt a slightly different view, where noise injection is introduced to enhance randomness for multi-scale stochastic variations. Different from the settings in BigGAN, they inject extra noise independent of latent inputs into different layers of the network without projection. Our theoretical analysis is mainly motivated by the success of noise injection used in StyleGAN (Karras et al., 2019a). Our proposed framework reveals that noise injection in StyleGAN is a kind of fuzzy reparameterization in Euclidean spaces, and we extends it into generic manifolds (section 4.3).

## 3 THE INTRINSIC DRAWBACKS OF TRADITIONAL GANS

### 3.1 OPTIMAL TRANSPORTATION AND DISCONTINUOUS GENERATOR

Traditional GANs with Wasserstein distance are equivalent to the optimal transportation problem, where the optimal generator is the optimal transportation map. However, there is rare chance for the optimal transportation map to be continuous, unless the support of Brenier potential is convex (Caffarelli, 1992). Considering that the Brenier potential of Wasserstein GAN is determined by the real data distribution and the inverse map of the generator, it is highly unlikely that its support is convex. This means that the optimal generator will be discontinuous, which is a fatal limitation to the capacity of GANs. Based on that, Lei et al. (Lei et al., 2019a) further point out that traditional GANs will hardly converge or converge to one continuous branch of the target mapping, thus leading to mode collapse. They then propose to find the continuous Brenier potential instead of the discontinuous

transportation map. In the next paragraph, we show that this solution may not totally overcome the problem that traditional GANs encounter due to structural limitations of neural networks. Besides, it suffices to note that their analysis is built upon the Wasserstein distance, and may not be directly applied to the Jenson-Shannon divergence or KL divergence. We refer the readers to Lei et al. (2019a); Caffarelli (1992) for more detailed analysis.

## 3.2 ADVERSARIAL DIMENSION TRAP

In addition to the above discontinuity problem, another drawback is the relatively low dimension of latent spaces in GANs compared with the high variance of details in real-world data. Taking face images as an example, the hair, freckles, and wrinkles have extremely high degree of freedom, which make traditional GANs often fail to capture them. The repetitive application of non-invertible CNN blocks makes the situation even worse. Non-invertible CNN, which is a singular linear transformation, will drop the intrinsic dimensions of feature manifolds (Strang et al., 1993). So during the feedforward procedure of the generator, the dimensions of feature spaces will keep being dropped. Then it will have a high chance that the valid dimension of the input latent space is lower than that of the real data. The relatively lower dimension of the input latent space will then force the dimension of the support with respect to the distribution of generated images lower than that of the real data, as no smooth mappings increase the dimension. However, the discriminator, which measures the distance of these two distributions, will keep encouraging the generator to increase the dimension up to the same as the true data. This contradictory functionality, as we show in the theorem bellow, incurs severe punishment on the smoothness and invertibility of the generative model, which we refer as the adversarial dimension trap.

**Theorem 1.** [2] *For a deterministic GAN model and generator $G : \mathcal{Z} \to \mathcal{X}$, if the dimension of the input latent $\mathcal{Z}$ is lower than that of data manifold $\mathcal{X}$, then at least one of the two cases must stand:*

1. *the generator cannot be Lipschitz;*
2. *the generator fails to capture the data distribution and is unable to perform inversion. Namely, for an arbitrary point $x \in \mathcal{X}$, the possibility of $G^{-1}(x) = \emptyset$ is 1.*

The above theorem stands for a wide range of GAN loss functions, including Wasserstein divergence, Jenson-Shannon divergence, and other KL-divergence based losses. Notice that this theorem implies much worse situation than it states. For any open sphere $B$ in the data manifold $\mathcal{X}$, the generator restricted in the pre-image of $B$ also follows this theorem, which suggests bad properties of nearly every local neighborhood. This also suggests that the above consequences of Theorem 1 may both stand. As in some subsets, the generator may successfully capture the data distribution, while in some others, the generator may fail to do so.

The first issue in section 3.1 can be addressed by not learning the generator directly with continuous neural network components. We will show how our method addresses the second issue.

## 4 FUZZY REPARAMETERIZATION

The generator $G$ in the traditional GAN is a composite of sequential non-linear feature mappings, which can be denoted as $G(z) = f^k \circ f^{k-1} \circ \cdots \circ f^1(z)$, where $z \sim \mathcal{N}(0, 1)$ is the standard Gaussian. Each feature mapping, which is typically a single layer convolutional neural network (CNN) plus non-linear activations, carries out a certain purpose such as extracting multi-scale patterns, upsampling, or merging multi-head information. The whole network is then a deterministic mapping from the latent space $\mathcal{Z}$ to the image space $\mathcal{X}$. We propose to replace $f^i(x), 1 \le i \le k$, with

$$g^i(x) = \mu^i(x) + \sigma^i(x)\epsilon, \ \epsilon \sim \mathcal{N}(0, 1), \ x \in g^{i-1} \circ \cdots \circ g^1(\mathcal{Z}). \tag{1}$$

We call it as Fuzzy Reparameterization (FR) as it in fact learns fuzzy equivalence relation of the original features, and uses reparameterization to model the high-dimensional feature manifolds. We believe that this is the proper form of generalization of noise injection in StlyeGAN, and will show the reasons and benefits in the following sub-sections.

---

[2] As the common practice in the manifold learning community, our theorems and discussions are based on Riemannian manifolds. Proofs to all the theorems are included in the supplementary material.

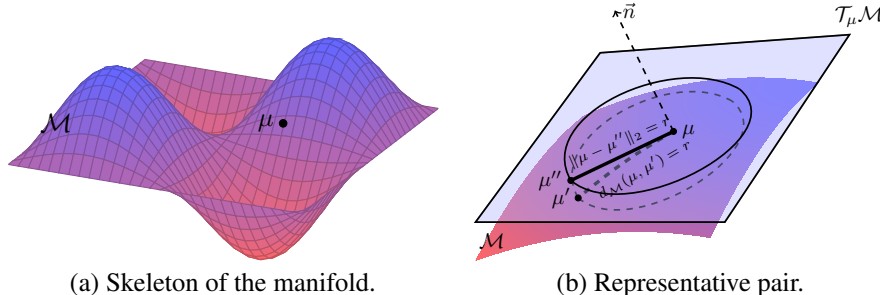

(a) Skeleton of the manifold.      (b) Representative pair.

Figure 2: Illustration of the skeleton set and representative pair. The blue curve in (a) is the skeleton. In (b), the dashed sphere in $\mathcal{M}$ is the geodesic ball, while the solid sphere in $\mathcal{T}_\mu\mathcal{M}$ is its projection onto the tangent space. The normal vector $\vec{n}$ determines the final affine transformation into the Euclidean space.

It is not hard to see that our proposed method can be viewed as the extension of the reparameterization trick in VAEs (Kingma & Welling, 2013). While the reparameterization trick in VAEs serves to a differentiable solution to learn through random variables and is only applied in the latent space, our method is a type of *deep* noise injection in feature maps of each layer to correct the defect in GAN architectures. Therefore, the purposes of using reparameterization in these two scenarios are different, thus leading to thoroughly different theories that are presented in the next sub-section.

## 4.1 HANDLING ADVERSARIAL DIMENSION TRAP WITH NOISE INJECTION

As Sard's theorem tells us (Petersen et al., 2006), the key to solve the adversarial dimension trap is to avoid mapping low-dimensional feature spaces into high-dimensional ones, which looks like a pyramid structure in the generator. However, we really need the pyramid structure in practice because the final output dimension of generated images is much larger than that of the input space. So the solution could be that, instead of mapping into the full feature spaces, we choose to map only onto the skeleton of the feature spaces and use random noise to fill up the remaining space. For a compact manifold, it is easy to find that the intrinsic dimension of the skeleton set can be arbitrarily low by applying Heine–Borel theorem to the skeleton (Rudin et al., 1964). By this way, the model can escape from the adversarial dimension trap.

Now we develop the idea in detail. The whole idea is based on approximating the manifold by the tangent polyhedron. Assume that the feature space $\mathcal{M}$ is a Riemannian manifold embedded in $\mathbf{R}^m$. Then for any point $\mu \in \mathcal{M}$, the local geometry induces a coordinate transformation from a small neighborhood of $\mu$ in $\mathcal{M}$ to its projection onto the tangent space $\mathcal{T}_\mu\mathcal{M}$ at $\mu$ by the following theorem.

**Theorem 2.** *Given Riemannian manifold $\mathcal{M}$ embedded in $\mathbf{R}^m$, for any point $\mu \in \mathcal{M}$, we let $\mathcal{T}_\mu\mathcal{M}$ denote the tangent space at $\mu$. Then the exponential map $Exp_\mu$ induces a smooth diffeomorphism from a Euclidean ball $B_{\mathcal{T}_\mu\mathcal{M}}(0, r)$ centered at $O$ to a geodesic ball $B_{\mathcal{M}}(\mu, r)$ centered at $\mu$ in $\mathcal{M}$. Thus $\{Exp_\mu^{-1}, B_{\mathcal{M}}(\mu, r), B_{\mathcal{T}_\mu\mathcal{M}}(0, r)\}$ forms a local coordinate system of $\mathcal{M}$ in $B_{\mathcal{M}}(\mu, r)$, which we call the normal coordinates. Thus we have*

$$B_{\mathcal{M}}(\mu, r) = Exp_\mu(B_{\mathcal{T}_\mu\mathcal{M}}(0, r)) = \{\tau : \tau = Exp_\mu(v), v \in B_{\mathcal{T}_\mu\mathcal{M}}(0, r)\}. \tag{2}$$

**Theorem 3.** *The differential of $Exp_\mu$ at the origin of $\mathcal{T}_\mu\mathcal{M}$ is identity $I$. Thus $Exp_\mu$ can be approximated by*

$$Exp_\mu(v) = \mu + Iv + o(\|v\|_2). \tag{3}$$

*Thus, if $r$ in equation (2) is small enough, we can approximate $B_{\mathcal{M}}(\mu, r)$ by*

$$B_{\mathcal{M}}(\mu, r) \approx \mu + IB_{\mathcal{T}_\mu\mathcal{M}}(0, r) = \{\tau : \tau = \mu + Iv, v \in B_{\mathcal{T}_\mu\mathcal{M}}(0, r)\}. \tag{4}$$

*Considering that $\mathcal{T}_\mu\mathcal{M}$ is an affine subspace of $\mathbf{R}^m$, the coordinates on $B_{\mathcal{T}_\mu\mathcal{M}}(0, r)$ admit an affine transformation into the coordinates on $\mathbf{R}^m$. Thus equation (4) can be written as*

$$B_{\mathcal{M}}(\mu, r) \approx \mu + IB_{\mathcal{T}_\mu\mathcal{M}}(0, r) = \{\tau : \tau = \mu + rT(\mu)\epsilon, \epsilon \in B(0, 1)\}. \tag{5}$$

We remind the readers that the linear component matrix $T(\mu)$ differs at different $\mu \in \mathcal{M}$ and is decided by the local geometry near $\mu$. In the above formula, $\mu$ defines the center point and $rT(\mu)$ defines the shape of the approximated neighbor. So we call them a representative pair of $B_{\mathcal{M}}(\mu, r)$.

Picking up a series of such representative pairs, which we refer as the skeleton set, we can construct a tangent polyhedron $\mathcal{H}$ of $\mathcal{M}$. Thus instead of trying to learn the feature manifold directly, we adopt a two-stage procedure. We first learn a map $f : x \mapsto [\mu(x), \sigma(x)]$ ($\sigma(x) \equiv rT(\mu(x))$) onto the skeleton set, then we use noise injection $g : x \mapsto \mu(x) + \sigma(x)\epsilon, \epsilon \sim \mathcal{U}(0, 1)$ (uniform distribution) to fill up the flesh of the feature space as shown in Figure 2.

However, the real world data often include fuzzy semantics. Even long range features could share some structural relations in common. It is unwise to model it with unsmooth architectures such as locally bounded sphere and uniform distribution. Thus we borrow the idea from fuzzy topology (Ling & Bo, 2003; Zhang & Zhang, 2005; Murali, 1989; Recasens, 2010) which is designed to address this issue. It is well known that for any distance metrics $d(\cdot, \cdot)$, $e^{-d(\mu, \cdot)}$ admits a fuzzy equivalence relation for points near $\mu$, which is similar with the density of Gaussian. The fuzzy equivalence relation can be viewed as a suitable smooth alternative to the sphere neighborhood $B_{\mathcal{M}}(\mu, r)$. Thus we replace the uniform distribution with unclipped Gaussian[3]. Under this settings, the first stage mapping in fact learns a fuzzy equivalence relation, while the second stage is a reparameterization technique. Notice that the skeleton set can have arbitrarily low dimension by Heine–Borel theorem. So the first-stage map can be smooth and well conditioned. For the second stage, we can show that it possesses a smooth property in expectation by the following theorem.

**Theorem 4.** *Given $f : x \mapsto [\mu(x), \sigma(x)]^T$, $f$ is locally Lipschitz and $\|\sigma\|_\infty = o(1)$. Define $g(x) \equiv \mu(x) + \sigma(x)\epsilon, \epsilon \sim \mathcal{N}(0, 1)$ (standard Gaussian). Then for any bounded set $U$, $\exists L > 0$, we have $\mathbf{E}[\|g(x) - g(y)\|_2] \leq L\|x - y\|_2 + o(1), \forall x, y \in U$. Namely, the principal component of $g$ is locally Lipschitz in expectation. Specifically, if the definition domain of $f$ is bounded, then the principal component of $g$ is globally Lipschitz in expectation.*

## 4.2 PROPERTIES OF NOISE INJECTION

As we have discussed, traditional GANs face two challenges: the discontinuous optimal generator and the adversarial dimension trap. Both of the two challenges will lead to an unsmooth generator. Theorem 1 also implies an unstable training procedure because the gradient explosion that may occur on the generator. Besides, the dimension reduction in GAN will make it hard to fit high-variance details as information keeps compressed along channels in the generator. With noise injection in the network of the generator, however, we can theoretically overcome such problems if the representative pairs are constructed properly to capture the local geometry. In this case, our model does not need to fit the discontinuous optimal transportation map, nor the image manifold with higher dimension than that the network architecture can handle. Thus the training procedure will not encourage the unsmooth generator, and can proceed more stably. Also, the extra noise can compensate the loss of information compression so as to capture high-variance details, which has been discussed and illustrated in StyleGAN (Karras et al., 2019a). We will evaluate the performance of our method from these aspects in section 5.

## 4.3 CHOICE OF $\mu(x)$ AND $\sigma(x)$

As $\mu$ stands for a particular point in the feature space, we simply model it by the traditional deep CNN architectures. $\sigma(x)$ is designed to fit the local geometry of $\mu(x)$. According to our theory, the local geometry should only admit minor differences from $\mu(x)$. Thus we believe that $\sigma(x)$ should be determined by the spatial and semantic information contained in $\mu(x)$, and should characterize the local variations of the spatial and semantic information. The deviation of pixel-wise sum along channels of feature maps in StyleGAN2 highlights the semantic variations like hair, parts of background, and silhouettes, as the standard deviation map over sampling instances shows in Fig. 1. This observation suggests that the sum along channels identifies the local semantics we expect to reveal. Thus it should be directly connected to $\sigma(x)$ we are pursuing here. For a given feature map $\mu = \mathbf{DCNN}(x)$ from the deep CNN, which is a specific point in the feature manifold, the sum along its channels is

$$\tilde{\mu}_{ijk} = \sum_{i=1}^{c} \mu_{ijk}, \tag{6}$$

---

[3]A detailed analysis about why unclipped Gaussian should be applied is offered in the supplementary material.

where $i$ enumerates all the $c$ feature maps of $\mu$, while $j, k$ enumerate the spatial footprint of $\mu$ in its $h$ rows and $w$ columns, respectively. The resulting $\tilde{\mu}$ is then a spatial semantic identifier, whose variation corresponds to the local semantic variation. We then normalize $\tilde{\mu}$ to obtain a spatial semantic coefficient matrix $s$ with

$$
\begin{aligned}
mean(\tilde{\mu}) &= \frac{1}{h \times w} \sum_{j=1}^{h} \sum_{k=1}^{w} \tilde{\mu}_{jk}, \\
s &= \tilde{\mu} - mean(\tilde{\mu}), \\
max(|s|) &= \max_{1 \leq j \leq h, 1 \leq k \leq w} |s_{jk}|, \\
s &= \frac{s}{max(|s|)}.
\end{aligned}
\tag{7}
$$

Recall that the standard deviation of $s$ over sampling instances highlights the local variance in semantics. Thus $s$ can be decomposed into two independent components: $s_m$ that corresponds to the main content of the output image, which is almost invariant under changes of injected noise; $s_v$ that is associated with the variance that is induced by the injected noise, and is nearly orthogonal to the main content. We assume that this decomposition can be attained by an affine transformation on $s$ such that

$$
s_d = A * s + b = s_m + s_v, s_v * \mu \approx \mathbf{0}, \tag{8}
$$

where $*$ denotes element-wise matrix multiplication, and $\mathbf{0}$ denotes the matrix whose all elements are zeros. To avoid numerical instability, we add $\mathbf{1}$ whose all elements are ones to the above decomposition, such that its condition number will not get exploded,

$$
\begin{aligned}
s' &= \alpha s_d + (1 - \alpha)\mathbf{1}, \\
\sigma &= \frac{s'}{\|s'\|_2}.
\end{aligned}
\tag{9}
$$

The regularized $s_m$ component is then used to enhance the main content in $\mu$, and the regularized $s_v$ component is then used to guide the variance of injected noise. The final output $o$ is then calculated as

$$
o = r\sigma * \mu + r\sigma * \epsilon, \epsilon \sim \mathcal{N}(0, 1). \tag{10}
$$

In the above procedure, $A, b, r$, and $\alpha$ are learnable parameters. Note that in the last equation, we do not need to decompose $s'$ into $s_v$ and $s_m$, as $s_v$ is designed to be nearly orthogonal to $\mu$, and $s_m$ is nearly invariant. Thus $\sigma * \mu$ will automatically drop the $s_v$ component, and $\sigma * \epsilon$ amounts to adding an invariant bias to the variance of injected noise. There are alternative forms for $\mu$ and $\sigma$ with respect to various GAN architectures. However, modeling $\mu$ by deep CNNs and deriving $\sigma$ through the spatial and semantic information of $\mu$ are universal for GANs, as they comply with our theorems. We further conduct ablation study to verify the effectiveness of the above procedure. The related results can be found in the supplementary material.

Using our formulation, noise injection in StyleGAN2 can be written as follows:

$$
\mu = \mathbf{DCNN}(x), o = \mu + r * \epsilon, \epsilon \sim \mathcal{N}(0, 1), \tag{11}
$$

where $r$ is a learnable *scalar* parameter. This can be viewed as a special case of our method, where $T(\mu)$ in (5) is set to identity. Under this settings, the local geometry is assumed to be everywhere identical among the feature manifold, which suggests a globally Euclidean structure. While our theory supports this simplification and specialization, our choice of $\mu(x)$ and $\sigma(x)$ can suit broader and more usual occasions, where the feature manifolds are non-Euclidean. We denote this fashion of noise injection as additive noise injection, and will extensively study its performance compared with our choice in the following section.

## 5 EXPERIMENT

We conduct experiments on benchmark datasets including FFHQ faces, LSUN objects, and CIFAR-10. The GAN models we use are the baseline DCGAN (Radford et al., 2015) (originally without noise injection) and the state-of-the-art StyleGAN2 (Karras et al., 2019b) (originally with additive noise injection). For StyleGAN2, we use config-e in the original paper due to that config-e achieves the best performance with respect to Path Perceptual Length (PPL) score. Besides, we apply the experimental settings from StyleGAN2.

Table 1: Comparison for different generator architectures.

| GAN arch | FFHQ | | LSUN-Church | |
|---|---|---|---|---|
| | PPL ($\downarrow$) | FID ($\downarrow$) | PPL ($\downarrow$) | FID ($\downarrow$) |
| DCGAN | 2.97 | 45.29 | 33.30 | 51.18 |
| DCGAN + Additive noise | 3.14 | 44.22 | 22.97 | 54.01 |
| DCGAN + FR (Ours) | **2.83** | **40.06** | **22.53** | **46.31** |
| Bald StyleGAN2 | 28.44 | 6.87 | 425.7 | 6.44 |
| StyleGAN2 | 16.20 | 7.29 | 123.6 | 6.80 |
| StyleGAN2-NoPathReg + FR (Ours) | 16.02 | **7.14** | 178.9 | **5.75** |
| StyleGAN2 + FR (Ours) | **13.05** | 7.31 | **119.5** | 6.86 |

FFHQ               LSUN-Church

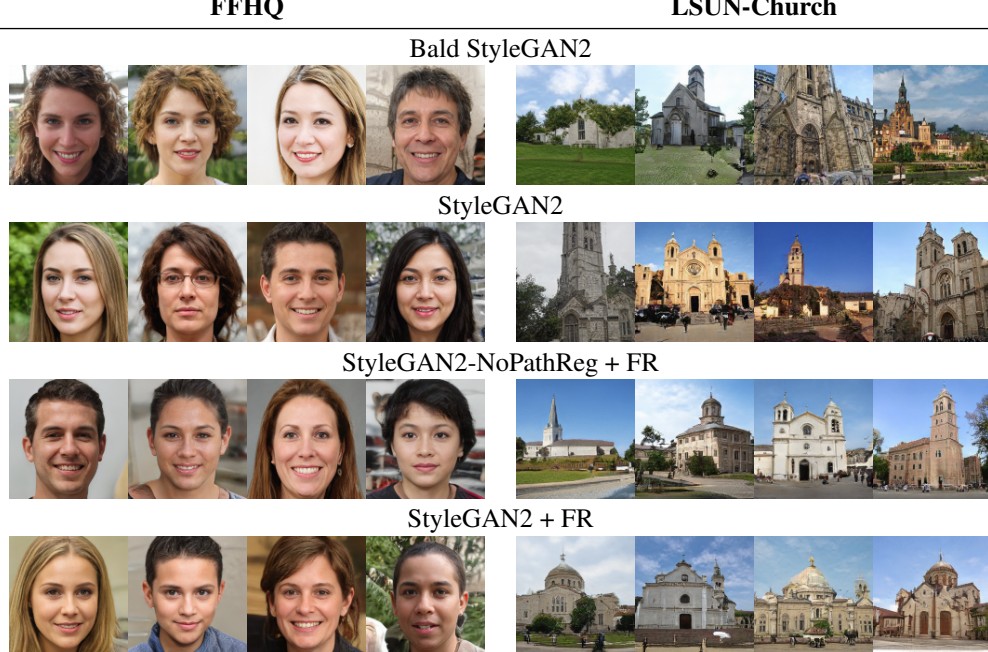

Figure 3: Synthesized images of different StyleGAN2-based models.

**Image synthesis.** PPL (Zhang et al., 2018) has been proven an effective metric for measuring structural consistency of generated images (Karras et al., 2019b). Considering its similarity to the expectation of the Lipschitz constant of the generator, it can also be viewed as a quantification of the smoothness of the generator. The path length regularizer is proposed in StyleGAN2 to improve generated image quality by explicitly regularizing the Jacobian of the generator with respect to the intermediate latent space. We first compare the noise injection methods with the bald StyleGAN2, which remove the additive noise injection and path length regularizer in StyleGAN2. As shown in Table 1, we can find that all types of noise injection significantly improve the PPL scores. It is worth noting that our method without path length regularizer can achieve comparable performance against the standard StyleGAN2 on the FFHQ dataset, and the performance can be further improved if combined with path length regularizer. Considering the extra GPU memory consuming of path length regularizer in training, we think that our method offers a computation-friendly alternative to StyleGAN2 as we observe smaller GPU memory occupation of our method throughout all the experiments. Another benefit is that our method accelerates the convergence to the optimal FID scores, as illustrated in Figure 4. This superior convergence can be explained with our theorem. The underlying reason is that our method offers an architecture that is more consistent with the intrinsic geometry of the feature space. Thus it is easier for the network to fit.

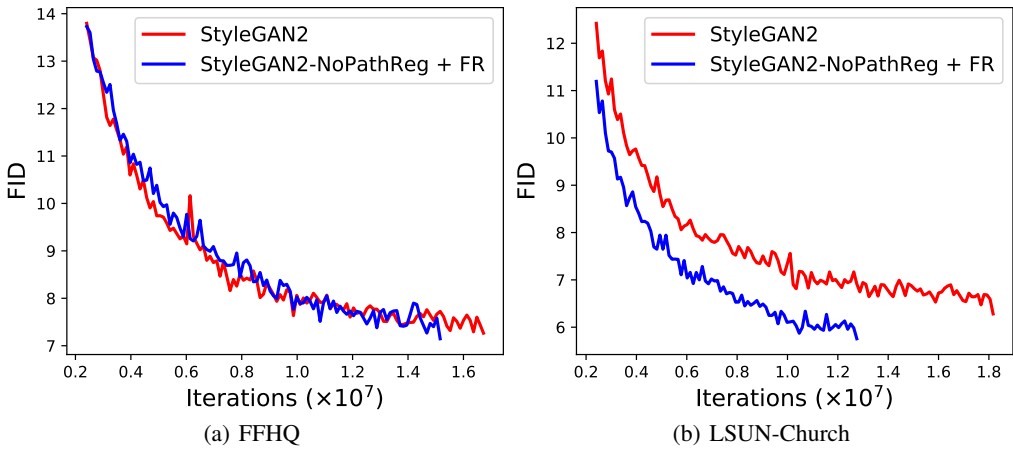

Figure 4: Comparison of FID curves. All curves are terminated by the optimal FIDs in 25M training iterations.

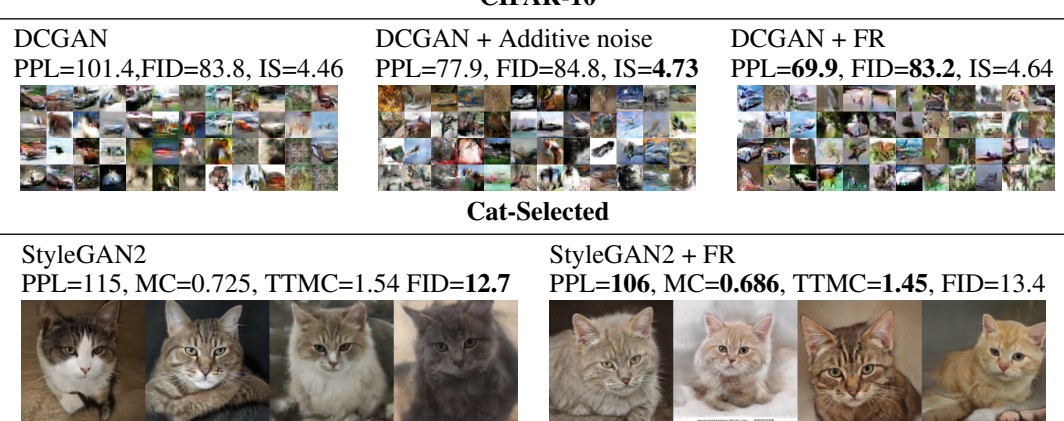

Figure 5: Image synthesis on CIFAR-10 and LSUN cats.

For the LSUN-Church dataset, we observe an obvious improvement in FID scores compared with StyleGAN2. We believe that this is because the LSUN-Church data are scene images and contain various semantics of multiple objects, which are hard to fit for the original StyleGAN2 that is more suitable for single object synthesis. So our FR architecture offers more degrees of freedom to the generator to fit the true distribution of the dataset. In all cases, our method is superior to StyleGAN2 in both PPL and FID scores. This proves that our noise injection method is more powerful than the one used in StyleGAN2. For DCGAN, as it does not possess the intermediate latent space, we cannot facilitate it with path length regularizer. So we only compare the additive noise injection with our FR method. Through all the cases we can find that our method achieves the best performance in PPL and FID scores.

Table 2: Conditions for different GAN architectures. MC and TTMC (Top Thousand Mean Condition) are mean condition and mean value of the largest 1000 conditions at 50000 randomly sampled points in the input space, respectively. The intermediate latent space is taken as the input space.

| GAN arch | FFHQ | | LSUN-Church | |
|---|---|---|---|---|
| | MC ($\downarrow$) | TTMC ($\downarrow$) | MC ($\downarrow$) | TTMC ($\downarrow$) |
| Bald StyleGAN2 | 0.943 | 2.81 | 2.31 | 6.31 |
| StyleGAN2 | 0.666 | 1.27 | 0.883 | 1.75 |
| StyleGAN2-NoPathReg + FR | 0.766 | 2.39 | 1.71 | 4.74 |
| StyleGAN2 + FR | **0.530** | **1.05** | **0.773** | **1.51** |

We also study whether our choice for $\mu(x)$ and $\sigma(x)$ can be applied to broader occasions. We further conduct experiments on a cat dataset which consists of 100 thousand selected images from 800 thousand LSUN-Cat images by PageRank algorithm (Zhou et al., 2004). For DCGAN, we conduct extra experiments on CIFAR-10 to test whether our method could succeed in multi-class image synthesis. The results are reported in Figure 5. We can see that our method still outperforms the compared methods in PPL scores and the FID scores are comparable, indicating that the proposed noise injection is more favorable of preserving structural consistency of generated images with real ones.

**Numerical stability.** As we have analyzed before, noise injection should be able to improve the numerical stability of GAN models. To evaluate it, we examine the condition number of different GAN architectures. The condition number of a given function $f$ is defines as Horn & Johnson (2013)

$$Cond(f) = \lim_{\delta \to 0} \sup_{\|\Delta x\| \leq \delta} \frac{\|f(x) - f(x + \Delta x)\|/\|f(x)\|}{\|\Delta x\|/\|x\|}. \tag{12}$$

It measures how sensitive a function is to changes or errors in the input. A function with a high condition number is said to be ill-conditioned. Considering the numerical infeasibility of the $\sup$ operator in the definition of condition number, we resort to the following alternative approach. We first sample a batch of 50000 pairs of $(Input, Perturbation)$ from the input distribution and the perturbation $\Delta x \sim \mathcal{N}(0, 1\text{e-}4)$, and then compute the corresponding condition numbers. We compute the mean value and the mean value of the largest 1000 values of these 50000 condition numbers as Mean Condition (**MC**) and Top Thousand Mean Condition (**TTMC**) respectively to evaluate the condition of GAN models. We report the results in Table 2, where we can find that noise injection significantly improves the condition of GAN models, and our proposed method dominates the performance.

**GAN inversion.** StyleGAN2 makes use of a latent style space that is capable of enabling controllable image modifications. This characteristic motivates us to study the image embedding capability of our method via GAN inversion algorithms (Abdal et al., 2019) as it may help further leverage the potential of GAN models. From the experiments, we find that the StyleGAN2 model is prone to work well for full-face, non-blocking human face images. For this type of images, we observe comparable performance for all the GAN architectures. We think that this is because those images are close to the 'mean' face of FFHQ dataset (Karras et al., 2019a), thus easy to learn for the StyleGAN-based models. For faces of large pose or partially occluded ones, the capacity of compared models differs significantly. Noise injection methods outperform the bald StyleGAN2 by a large margin, and our method achieves the best performance. The detailed implementation and results are reported in the supplementary material.

## 6    Conclusion

In this paper, we propose a theoretical framework to explain the effect of noise injection technique in GANs. We prove that the generator can easily encounter difficulty of unsmoothness, and noise injection is an effective approach to addressing this issue. Based on our theoretical framework, we also derive a more proper formulation for noise injection. We conduct experiments on various datasets to confirm its validity. Despite the superiority compared with the existing methods, however, it is still unclear whether our formulation is optimal and universal for different networks. In future work, we will further investigate the realization of noise injection, and attempt to find more powerful way to characterize local geometries of feature spaces.

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

# Appendices

## A    PROOF TO THEOREMS

### A.1    THEOREM 1

*Proof.* Denote the dimensions of $G(\mathcal{Z})$ and $\mathcal{X}$ as $d_G$ and $d_\mathcal{X}$, respectively. There are two possible cases for $G$: $d_G$ is lower than $d_\mathcal{X}$, or $d_G$ is higher than or equal to $d_\mathcal{X}$.

For the first case, a direct consequence is that, for almost all points in $\mathcal{X}$, there are no pre-images under $G$. This means that for an arbitrary point $x \in \mathcal{X}$, the possibility of $G^{-1}(x) = \emptyset$ is 1, as $\{x \in \mathcal{X} : G^{-1}(x) \neq \emptyset\} \subset G(\mathcal{Z}) \cap \mathcal{X}$, which is a zero measure set in $\mathcal{X}$. This also implies that the generator is unable to perform inversion. Another consequence is that, the generated distribution $P_g$ can never get aligned with real data distribution $P_r$. Namely, the distance between $P_r$ and $P_g$ cannot be zero for arbitrary distance metrics. For the KL divergence, the distance will even approach infinity.

For the second case, $d_G \geq d_{\mathcal{X}} > d_{\mathcal{Z}}$. We simply show that a Lipschitz-continuous function cannot map zero measure set into positive measure set. Specifically, the image of low dimensional space of a Lipschitz-continuous function has measure zero. Thus if $d_G \geq d_{\mathcal{X}}$, $G$ cannot be Lipschitz.

Now we prove our claim.

Suppose that $f : \mathbb{R}^n \to \mathbb{R}^m, n < m$, $f$ is Lipschitz with Lipschitz constant $L$. We show that $f(\mathbb{R}^n)$ has measure zero in $\mathbb{R}^m$. As $\mathbb{R}^n$ is a zero measure subset of $\mathbb{R}^m$, by the Kirszbraun theorem (Deimling, 2010), $f$ has an extension to a Lipschitz function of the same Lipschitz constant on $\mathbb{R}^m$. For convenience, we still denote the extension as $f$. Then the problem reduces to proving that $f$ maps zero measure set to zero measure set. For every $\epsilon > 0$, we can find countable union of balls $\{B_k\}_k$ of radius $r_k$ such that $\mathbb{R}^n \subset \cup_k B_k$ and $\sum_k m(B_k) < \epsilon$ in $\mathbb{R}^m$, where $m(\cdot)$ is the Lebesgue measure in $\mathbb{R}^m$. But $f(B_k)$ is contained in a ball with radius $Lr_k$. Thus we have $m(f(\mathbf{R^n})) \leq L^m \sum_k m(B_k) < L^m \epsilon$, which means that it is a zero measure set in $\mathbb{R}^m$. For the mapping between manifolds, using the chart system can turn it into the case we analyze above, which completes our proof. $\qquad\square$

We want to remind the readers that, even if the generator suits one of the cases in Theorem 1, the other case can still occur. For example, $G$ could succeed in capturing the distribution of certain parts of the real data, while it may fail in the other parts. Then for the pre-image of those successfully captured data, the generator will not have finite Lipschitz constant.

## A.2  THEOREMS 2 & 3

Theorems 2 & 3 are classical conclusions in Riemannian manifold. We refer readers to section 5.5 of the book written by Petersen et al. (2006) for detailed proofs and illustration.

## A.3  THEOREM 4

*Proof.*

$$\mathbf{E}[\|g(x) - g(y)\|_2] \leq \|\mu(x) - \mu(y)\|_2 + \mathbf{E}[\|\sigma(x)\epsilon - \sigma(y)\delta\|_2] \tag{13}$$
$$\leq L_\mu \|x - y\|_2 + 2C\|\sigma\|_\infty \leq L_\mu \|x - y\|_2 + o(1), \tag{14}$$

where $C$ is a constant related to the dimension of the image space of $\sigma$ and $L_\mu$ is the Lipschitz constant of $\mu$. $\qquad\square$

## B  WHY GAUSSIAN DISTRIBUTION?

We first introduce the notion of fuzzy equivalence relations.

**Definition 1.** *A t-norm is a function* $T : [0,1] \times [0,1] \to [0,1]$ *which satisfies the following properties:*

  1. *Commutativity:* $T(a,b) = T(b,a)$.

  2. *Monotonicity:* $T(a,b) \leq T(c,d)$, *if* $a \leq c$ *and* $b \leq d$.

  3. *Associativity:* $T(a, T(b,c)) = T(T(a,b), c)$.

  4. *The number 1 acts as identity element:* $T(a,1) = a$.

**Definition 2.** *Given a t-norm* $T$, *a* $T$-*equivalence relation on a set* $X$ *is a fuzzy relation* $E$ *on* $X$ *and satisfies the following conditions:*

  1. $E(x,x) = 1, \forall x \in X$ *(Reflexivity)*.

  2. $E(x,y) = E(y,x), \forall x, y \in X$ *(Symmetry)*.

  3. $T(E(x,y), E(y,z)) \leq E(x,z) \forall x, y, z \in X$ *(T-transitivity)*.

Then it is easy to check that $T(x,y) = xy$ is a t-norm, and $E(x,y) = e^{-d(x,y)}$ is a $T$-equivalence for any distance metric $d$ on $X$, as

$$T(E(x,y), E(y,z)) = e^{-(d(x,y) + d(y,z))} \leq e^{-d(x,z)} = E(x,z).$$

Table 3: GPU environments for all experiments in this work.

| Experiment | Environment |
|---|---|
| StyleGAN2 based GAN model training | 8 NVIDIA Tesla V100-SXM2-16GB GPUs (DGX-1 station) |
| DCGAN based GAN model training | 4 TITAN Xp GPUs |
| Metrics measurement | 8 GeForce GTX 1080Ti GPUs |
| GAN inversion | 1 TITAN Xp GPU |

Considering that we want to contain the fuzzy semantics of real world data in our local geometries of feature manifolds, a natural solution will be that we sample points from the local neighborhood of $\mu$ with different densities on behalf of different strength of semantic relations with $\mu$. Points with stronger semantic relations will have larger densities to be sampled. A good framework to model this process is the fuzzy equivalence relations we mention above, where the degrees of membership $E$ are used as the sampling density. However, our expansion of the exponential map $Exp_\mu$ carries an error term of $o(\|v\|_2)$. We certainly do not want the local error to be out of control, and we also wish to constrain the sampling locally. Thus we accelerate the decrease of density when points depart from the center $\mu$, and constrain the integral of $E$ to be identity, which turns $E$ to the density of standard Gaussian.

## C  DATASETS

**FFHQ**  Flickr-Faces-HQ (FFHQ) (Karras et al., 2019a) is a high-quality image dataset of human faces, originally created as a benchmark data for generative adversarial networks (GANs). The dataset consists of 70,000 high-quality PNG images and contains considerable variations in terms of age, pose, expression, hair style, ethnicity and image backgrounds. It also covers diverse accessories such as eyeglasses, sunglasses, hats, etc.

**LSUN-Church and Cat-Selected**  LSUN-Church is the church outdoor category of LSUN dataset (Yu et al., 2015), which consists of 126 thousand church images of various styles. Cat-Selected contains 100 thousand cat images selected by ranking algorithm (Zhou et al., 2004) from the LSUN cat category. The plausibility of using PageRank to rank data was analyzed in Zhou et al. (2004). We also used the algorithm presented in Zhao & Tang (2009) to construct the graph from the cat data.

**CIFAR-10**  The CIFAR-10 dataset (Krizhevsky et al., 2009) consists of 60,000 images of size 32x32. There are all 10 classes and 6000 images per class. There are 50,000 training images and 10,000 test images.

## D  IMPLEMENTATION DETAILS

### D.1  MODELS

We illustrate the generator architectures of StyleGAN2 based methods in Figure 6. For all those models, the discriminators share the same architecture as the original StyleGAN2. The generator architecture of DCGAN based methods are illustrated in Figure 7. For all those models, the discriminators share the same architecture as the original DCGAN.

## E  EXPERIMENT ENVIRONMENT

All experiments are carried out by TensorFlow 1.14 and Python 3.6 with CUDA Version 10.2 and NVIDIA-SMI 440.64.00. We basically build our code upon the framework of NVIDIA official StyleGAN2 code, which is available at `https://github.com/NVlabs/stylegan2`. We use a variety of servers to run the experiments as reported in Table 3.

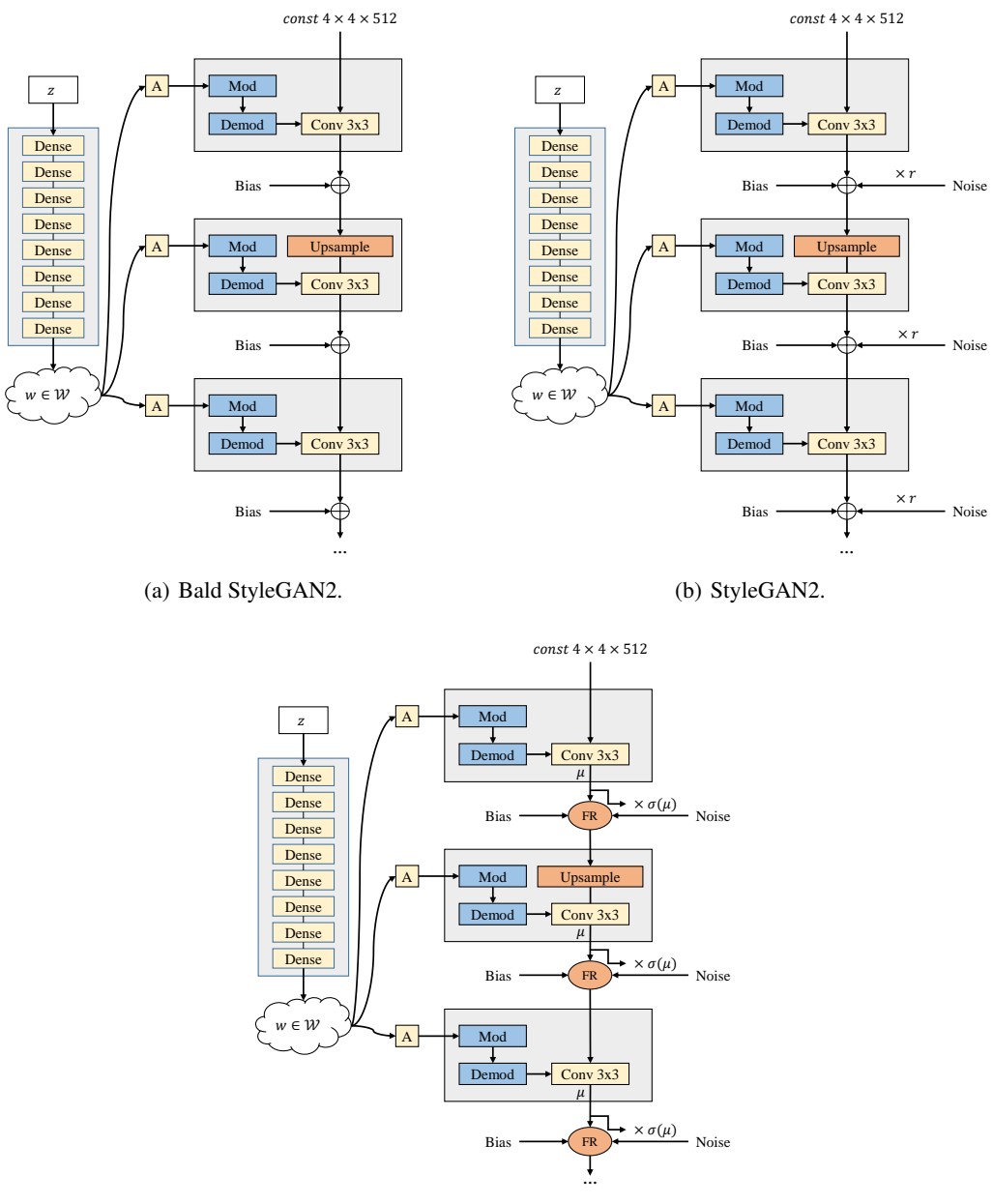

Figure 6: Generator architectures of StyleGAN2 based models. (a) The generator of bald StyleGAN2. (b) The generator of StyleGAN2. (c) The generator of StyleGAN2 + FR and StyleGAN2-NoPathReg + FR. 'Mod' and 'Demod' denote the weight demodulation method proposed in section 2.2 of StyleGAN2 (Karras et al., 2019b). $A$ denotes a learned affine transformation from the intermediate latent space $\mathcal{W}$.

## F   IMAGE ENCODING AND GAN INVERSION

From a mathematical perspective, a well behaved generator should be easily invertible. In the last section, we have shown that our method is well conditioned, which implies that it could be easily invertible. We adopt the methods in Image2StyleGAN (Abdal et al., 2019) to perform GAN inversion and compare the mean square error and perceptual loss on a manually collected dataset of 20 images. The images are shown in Figure 8 and the quantitative results are provided in Table 4. For our FR

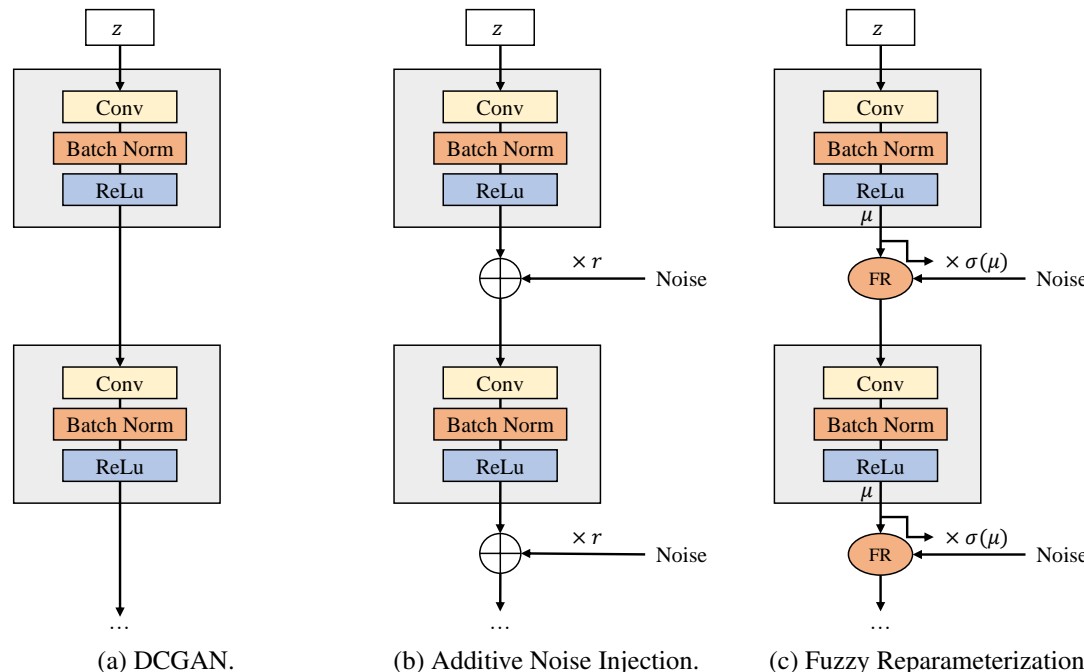

(a) DCGAN.  (b) Additive Noise Injection.  (c) Fuzzy Reparameterization.

Figure 7: Generator architecture of DCGAN based models. (a) The generator of DCGAN. (b) The generator of DCGAN + Additive Noise. (c) The generator of DCGAN + FR.

Table 4: Image inversion metrics for different StyleGAN2 based models. The perceptual loss is the mean square distance of VGG16 features between the original and projected images as in Abdal et al. (2019)

| GAN arch | Overall | | Hard Cases | |
|---|---|---|---|---|
| | MSE ($\downarrow$) | Perceptual Loss ($\downarrow$) | MSE ($\downarrow$) | Perceptual Loss ($\downarrow$) |
| Bald StyleGAN2 | 1.34 | 5.42 | 2.86 | 11.34 |
| StyleGAN2 | 1.24 | 4.86 | 2.58 | 9.82 |
| StyleGAN2-NoPathReg + FR | 1.24 | 5.11 | 2.70 | 10.49 |
| StyleGAN2 + FR | **1.13** | **4.52** | **2.23** | **8.47** |

methods, we further optimize the $\alpha$ parameter in Eq. 7 in section 4.3, which fine-tunes the local geometries of the network to suit the new images that might not be settled in the model. Considering that $\alpha$ is limited to $[0, 1]$, we use $\frac{(\alpha^*)^t}{(\alpha^*)^t + (1-\alpha^*)^t}$ to replace the original $\alpha$ and optimize $t$. The initial value of $t$ is set to 1.0 and $\alpha^*$ is constant with the same value as $\alpha$ in the converged FR models.

During the experiments, we find that the StyleGAN2 model is prone to work well for full-face, non-blocking human face images. For this type of images (which we refer as regular case in Figure 9), we observe comparable performance for all the GAN architectures. We think that this is because those images are closed to the 'mean' face of FFHQ dataset (Karras et al., 2019a), thus easy to learn for the StyleGAN based models. For faces of large pose or partially blocked ones, the capacity of different models differs significantly. Noise injection methods outperform the bald StyleGAN2 by a large margin, and our method achieves the best performance.

# G ABLATION STUDY OF FR

In Tab. 5, we perform the ablation study of the proposed FR method on the FFHQ dataset. We test 5 different choices of FR implementation and compare their FID and PPL scores after convergence.

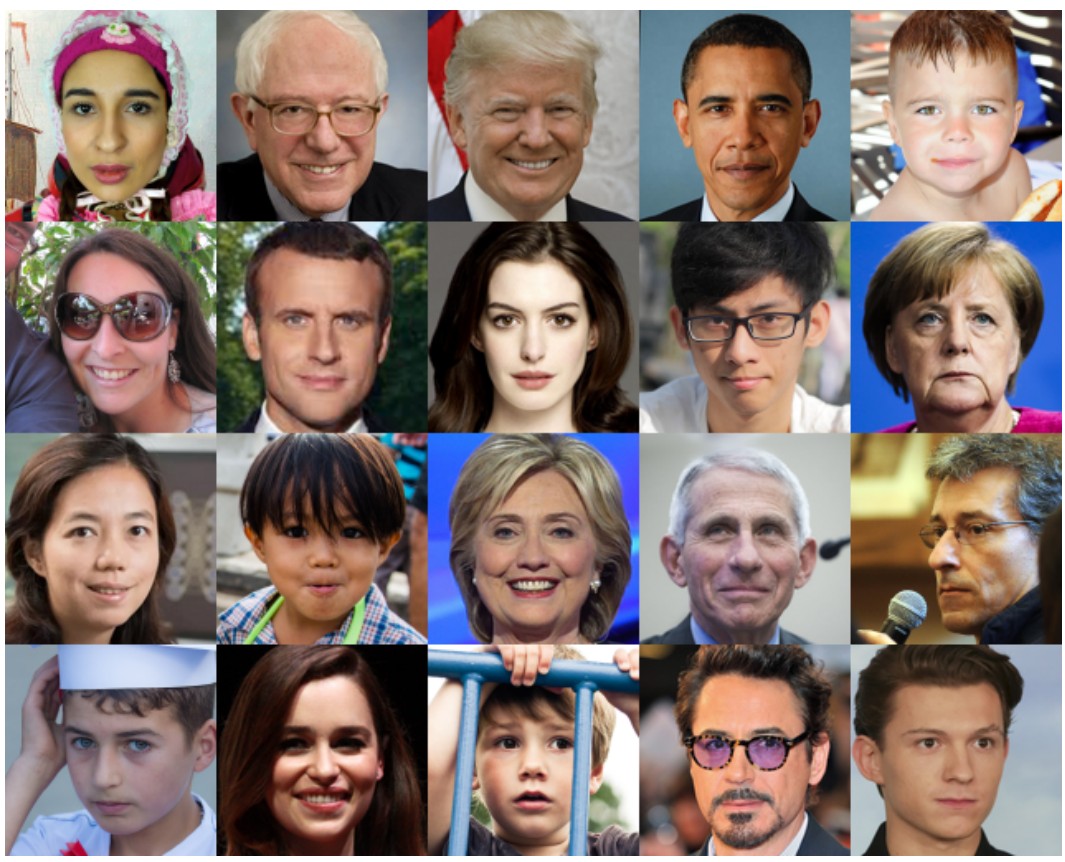

Figure 8: Manually collected 20 images for GAN inversion.

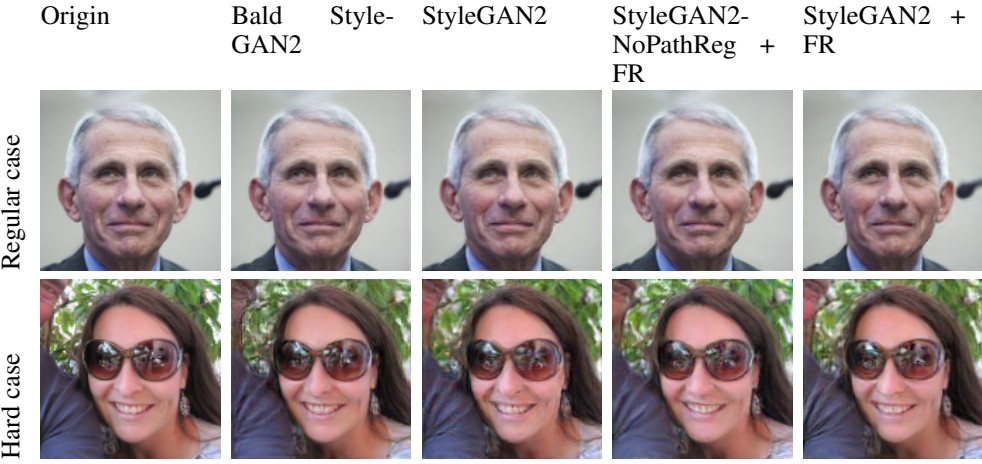

Figure 9: Projected Images to the intermediate latent spaces of StyleGAN2 based models.

1. No normalization: in this setting we remove the normalization of $\tilde{\mu}$ in Eq. 7, and use the unnormalized $\tilde{\mu}$ to replace $s$ in the following equations. The network comes to a minimum FID of 23.77 after training on 1323 thousand images, and then quickly falls into mode collapse after that.

Table 5: Ablation study of different noise injection methods on FFHQ. The zero values of PPL scores in the first two methods suggest mode collapse.

| Method | FID | PPL |
|---|---|---|
| No normalization | 628.94 | 0 |
| No stabilization | 184.30 | 0 |
| No decomposition | **6.48** | 18.78 |
| CNN | 22.54 | 14.53 |
| FR | 7.31 | **13.05** |

2. No stabilization: in this setting we remove the stabilization technique in Eq. 9. The network comes to a minimum FID of 50.27 after training on 963 thousand images, and then quickly falls to mode collapse after that.

3. No decomposition: in this setting we remove the decomposition in Eq. 8. The network successfully converges, but admits a large PPL score.

4. CNN: in this setting we use a convolutional neural network to replace the procedure that we get $\sigma$ in section 4.3. Namely, we take $\sigma = \mathbf{CNN}(\mu)$. The network successfully converges, but admits a very large FID score.

The zero PPL scores in 'No normalization' and 'No stabilization' suggest that the generator output is invariant to small perturbations, which means mode collapse. We can find that the stabilization and normalization in the FR implementation in section 4.3 is necessary for the network to avoid numerical instability and mode collapse. The implementation of FR method reaches the best performance in PPL score and comparable performance against the 'no decomposition' method in FID score. As analyzed in StyleGAN (Karras et al., 2019a) and StyleGAN2 (Karras et al., 2019b), for high fidelity images, PPL is more convincing than the FID score in measuring the synthesis quality. Therefore, the FR implementation is the best among these methods.

