# OpenReview forum: "On Noise Injection in Generative Adversarial Networks"
_ICLR.cc/2021/Conference — Reject_

### Official Review · AnonReviewer1 · 2020-10-20
**This work explains the properties and proposes a novel form of noise injection in GANs to reduce the adversarial dimension trap problem. Some issues in experiments need to be addressed.**

**Rating:** 6
**Confidence:** 3

**Review:**

Pros:

This work introduces the problem of adversarial dimension trap, which leads to punishment on the smoothness and invertibility of GANs.

This work proposes to learn fuzzy equivalence relation of the features and uses reparameterization trick to model the high-dimensional feature manifolds.

A novel form of noise injection is proposed to overcome the adversarial dimension trap. Prior noise injection methods can be explained as a special case with certain hyper-parameters. This method is universal for the families of GANs, including WGAN, DCGAN, etc.

Experiments on three datasets are conducted. The results of both image synthesis and GAN inversion are desirable with plausible texture details.


Cons:

My major concern is about the experiments. In Table 1, it seems that the reported FID and PPL differ from the scores reported by StyleGAN2 [1]. For the FFHQ dataset, [1] report that the FID is 3.31 (config E), but the FID of the baseline in this paper is 7.14 (the same setting). Such a huge discrepancy is wired. Recall that [1] improve the FID from 4.40 (StyleGAN v1) to 2.84, but the baseline, which should be the same, performs even worse than StyleGAN v1. The authors need to explain these contradictions.

The differences between the PPL scores reported in this paper and [1] are even more significant. I notice it is 13.05 (the best in this paper) versus 122.5 (the best in [1]). Why is the PPL about ten times better (even without the proposed method)?

The authors need to provide more details to explain how they calculated the FIDs and PPL. It seems that the authors calculate these scores in a non-standard manner. Besides, I suggest the author evaluate the Precision and Recall [2] on FFHQ. I wonder whether these metrics will be consistent.

[1] Karras, Tero, Samuli Laine, Miika Aittala, Janne Hellsten, Jaakko Lehtinen, and Timo Aila. "Analyzing and improving the image quality of stylegan." CVPR, pp. 8110-8119. 2020.

[2] Kynkäänniemi, Tuomas, Tero Karras, Samuli Laine, Jaakko Lehtinen, and Timo Aila. "Improved precision and recall metric for assessing generative models." NeurIPS, pp. 3927-3936. 2019.

---

> ### Author Response · Authors · 2020-11-18
> **To Reviewer #1**
>
> To Reviewer #1:
> We use the exact implementation in the StyleGAN2 official repository. We use the same experiment settings for all models. The reviewer may note that in this paper all experiments are conducted on 128 resolution images, thus the FID scores are different from those in the StyleGAN2 paper that uses 1024 resolution to compute FID.
>
> The resolution issue further influences the PPL score. The network depth is a primary factor in the PPL score. Recall its definition in the StyleGAN paper. the PPL score is some kind of ‘gradient norm’ with respect to the VGG network. The chain rule then implies that it will increase fast as network gets deeper. The StyleGAN2 for 1024 resolution images contains 12 more convolution layers than our 128 setting. As each convolution layer is further normalized by weight demodulation, they all maintain non-negligible gradient norms. The reviewer may also note that in Tab. 1 of StyleGAN2, the application of large networks brings a significant growth to the PPL score of StyleGAN2 on FFHQ (while the proposed method improves PPL though it introduces more parameters). These issues together result in a much larger base score of PPL. For LSUN church, the StyleGAN2 paper uses 256 resolution images, which is   closer to our 128 setting, thus the PPL score is also closer to ours.
>
> A similar difference of PPL is shown in the NIPS2019 workshop paper “Conditional Image Sampling by Deep Automodulators”, Ari Heljakka et. al. They measure the PPL score of StyleGAN on 256 resolution CelebA-HQ dataset in Tab. 1. The PPL score is 50.08, which also admits large margin from the score on 1024 resolution FFHQ.
>
> For recall and precision on 128 resolution FFHQ images, the plain StyleGAN2 has 0.6737 Precision score and 0.3772 Recall score; StyleGAN2 has 0.6831 Precision score and 0.3242 Recall score. The proposed FR method has 0.6670 Precison score and 0.3544 Recall score.

---

### Official Review · AnonReviewer4 · 2020-10-28
**Could contain useful bits but not ready for publication (but see updated part)**

**Rating:** 6
**Confidence:** 3

**Review:**

The paper analyzes the theoretical properties of noise injection in StyleGAN-like networks, and proposes an extension to in particular to StyleGAN2 that results in somewhat improved metric scores. Unfortunately the paper is rather confusingly written and hard to follow. To highlight what I mean, I will try to paraphrase my understanding of the paper in the following.

The theoretical treatment begins by framing the problem around optimal transport, but later seems to mostly drop this viewpoint. While the cited OT/GAN work presents interesting and relevant viewpoints about the difficulties in GAN training, I am not sure if it is particularly more relevant here than any number of other theoretical works. It may be noted that StyleGAN and DCGAN are not even formulated as to minimize a Wasserstein divergence.

The paper then coins a term "adversarial dimension trap", which I am not exactly sure why this terminology was chosen. The gist of the observation seems to be mostly well known, i.e. the generator can only cover a zero-measure region of the data space whereas the data is more spread out. That said, I am not thoroughly familiar with previous theoretical work on GANs and the particular formulation here may be novel. The paper then introduces a fairly general form of stochastic noise injection into the network layers and calls this fuzzy reparametrization. Here some connections to are drawn to "fuzzy equivalence relations" which (apparently?) are an existing concept, however as far as I see there is no citation to discuss these and little insight is given about why this is relevant.

Then, the key theory is developed. If I understand correctly, the key idea here is that the zero-measure manifold is "puffed up" with random distributions centered around points on it, with spread that depends on the point. This makes sense as a principle but overall I am confused about whether something fundamental was discrovered or proved, or whether this is just an introduction to the reasoning behind the practical algorithms.

This then leads to a proposal of a practical algorithm. If I understand correctly, it basically generalizes the StyleGAN noise injection in such a way that not only the mean, but also the stdev of each feature is predicted by the network. This plausibly allows for more flexibility. Unfortunately the description here is again rather confusing. Apparently formulas 6-8 are not really equations, but rather some kind of imperative pseudocode with variable assignments. It would be better to spell this out as an algorithm listing. As for the content of these formulas, I am not sure if I understand what the operations or the reasoning behind them is. Why the pixsum here? Apparently it produces a single value per feature map? After this there seems that these numbers are transformed by some global matrix(?) A and bias b, however I'm not sure what the convex combination with a matrix(?) I means here given that the first half of the formula is a vector (?). Then the result seems to be normalized again (why?) And finally the means are transformed by this standard deviation? There may well be good reasons to use these steps, but they are not explained so it ends up looking like an arbitrary heuristic. Here it would be important to make a strong connection to the insights derived from theory.

The presentation is further made confusing by the language. I understand that the authors may not be native speakers, but the readability is much below the usual standard of ICLR papers and the paper would benefit from improving this.

As for the results, it does appear that there is some improvement in some of the metrics, and the proposed method may in principle be useful. It is not hard to believe that adding some extra flexibility to the noise injection might improve the results, at least in a limited number of scenarios. In this sense the paper may be on to something.

What is the meaning of using PageRank to reduce the number of LSUN-Cat images? How is PageRank related to choosing images and what's the difference between that and just taking the first 100k pictures in the set? And for that matter, I am not sure if we learn anything from randomly limiting the set to 100k images, when we don't know how it worked for the full set. For the inversion experiments, the table in the appendix does show improvement and this may well be the case. In figure 9, though, it's hard to see much of a difference between any of the methods, perhaps in part because the images shown are very low resolution and do not correspond to anywhere near the SG2 output image size -- any differences in details are completely hidden by this.

The architecture figures 6-7 are unnecessarily low detail. They contain a black box "FR" node precisely at the place where you'd want to know more. Perhaps this node could be expanded into its own architecture diagram as well, given that there is no shortage of space in the appendix.

In summary the paper might contain useful bits -- this is somewhat hard to judge -- but whether or not that is the case, it is not in an acceptable condition without some significant rewriting, and I would recommend rejection at this time.

_UPDATE AFTER REBUTTAL_

The authors have improved the paper somewhat by expanding and clarifying the discussion on some key parts. While I think there is still much room for improvement in the paper, the general consensus seems to be towards acceptance. I will not oppose if that is the decision, and have increased my score accordingly. However I remain very borderline and I am not sure if I am fully convinced by all the claims.

One specific issue: I think the authors should make it more clear in the paper that the experiments are done in 128 pixel resolution, in light of R1's questions. It is important that the reader be aware of this, as the noise inputs arguably become much more important in high resolutions where there is more stochastic detail. I personally did not realize this when writing my review, and now wonder how the results would be at e.g. 256 or 512 resolution. If possible I would suggest the authors still run such experiments. This also probably explains my comment above on the lack of apparent visual differences in inversion results.

---

> ### Author Response · Authors · 2020-11-18
> **To Reviewer #4**
>
> To Reviewer #4:
>
> Concern1: Connection with the cited OT/GAN works.
>
> The idea of adversarial dimension trap is in fact a generalization of the discontinuity of optimal transportation maps. The cited OT/GAN works majorly concern the cases with a Wasserstein distance. The Jenson-Shannon divergence and other loss functions are widely applied in the GAN community, which motivates us to generalize it to broader occasions without specification of a certain loss function. In Appendix A.1, we actually demonstrate the adversarial dimension trap for three different loss functions, the Wasserstein distance, Jenson-Shannon divergence, and KL divergence. Our theory is not limited to any certain form of loss functions. We are obligated to cite and emphasize the cited OT/GAN works, because they may be the first to raise this concern of discontinuous generator from a rather solid Riemannian geometry perspective.
>
> Concern2: Term of ''Adversarial dimension trap’’
>
> The reviewer may misunderstand the key idea of Theorem 1. It does not mean that the generator can only learn a zero-measure set. It means that the discriminator will severely punish the intention of the generator to only learn a low-dimensional zero-measure set of the data manifold, which results in an unsmooth generator (the first case in Theorem 1), or a totally disabled discriminator and ill-conditioned generator (the second case in Theorem 1).  As it is a consequence of the adversarial game between generator and discriminator, we call it ''adversarial dimension trap''. To the best of our knowledge, this issue has not yet got widely concerned.
>
> Concern3: Citations to the fuzzy equivalence class.
>
> There are citations to the works of fuzzy topology in lines 3-4 of page 5. Please check it.
>
> Concern4: Whether something fundamental was discovered or proved.
>
> Theorems 2&3 PROVE that a Riemannian manifold can be well approximated by the form of noise injection method, and clearly state the meanings of different parts of noise injection. The noise injection should be small in order to obtain a globally first-order approximation. The variance matrix of injected noise stands for the local geodesic chart of the manifold. These are non-trivial results as noise injection itself is not included in the input of generator, and we do not control it when sampling images. It is very important to make sure that noise injection will not damage the good learning potential of the generator. Theorem 4 further proves that the noise injected network will admit a Lipschitz continuity locally. This confirms that the proposed method can address the adversarial dimension trap.
>
> Concern 5. Eq. 6-8.
>
> The major motivation behind eq. 6-8 is inspired by the deviation map in Fig. 1. The variance (sigma) of proposed noise injection method should model the local variance in the data manifold, which in its semantic meaning, is the detailed part of synthesis images, such as hair, parts of background, and silhouettes. The standard deviation map in Fig. 1 visualizes the deviation of sum of channels in the feature maps of StyleGAN blocks, which exactly corresponds to the semantic meaning of sigma. “A” is a learnable matrix which element-wisely controls the weight of contributions from each pixel of the feature map. “b”  is a learnable bias matrix to further adjust the bias of sigma. “r” is a scalar which controls the volume of injected noise as suggested in Theorems 2&3&4 to maintain locally Lipschitz property. Alpha is also a learnable scalar to explicitly regularize the structure of noise injection distribution and make sure that sigma is not ill-conditioned by adding an identity matrix to it., The matrices A and b in fact serve as a spatial attention enhancer to adjust the semantic attention in sigma. The proposed noise injection method is applied in each block of the StyleGAN2 network, thus resulting in a hierarchical adjustment to the detail information in the final outputs.
>
>
> Concern 6. Limited number of scenarios.
>
> The FFHQ, LSUN datasets are benchmarks for style-based GAN models. We follow the state-of-the-art papers to conduct the experiment.
>
> Concern 7. PageRank in LSUN Cat.
>
> The full LSUN-Cat dataset contains over 1 million images. The full-scale training of StyleGAN2 on it for ONE time will require 8 16G V100 GPUs for more than one week. We cannot afford such time-consuming with our limited server facilities. Thus, we have to select a small subset of it. Considering that the data variance in LSUN-Cat is considerable, we use PageRank to select a most compact subset to use, which can guarantee the success of GAN models for such data.

---

### Official Review · AnonReviewer2 · 2020-10-28
**Good contribution to the GAN research field**

**Rating:** 7
**Confidence:** 4

**Review:**

In this paper the authors highlight two major drawbacks of GANs. 1) The optimal Generator is discontinuous and 2) the 'adversarial dimension trap' caused by the relatively lower dimension of the latent space compared to the real-world data which makes the generator not Lipschitz and/or the generator fails to capture the real-world data distribution and is not invertible.
Both issues could lead to an unsmooth generator.

Secondly, the authors provide a form of generalization of noise injection in GANs called fuzzy reparameterization, which leads to a solution by letting the generator map onto an arbitrarily low dimensional skeleton of the feature spaces and filling up the remaining space with random noise. Therefore, the difference to the latent space dimensionality is minimized addressing issue 2). The solution consists of two stages, first a map from feature space onto the skeleton set is learned followed by noise injection adapted to the local geometry of the orginal feature manifold.

Experiments:

Experiments were done on FFHQ faces, LSUN objects, and CIFAR-10 datasets with models DCGAN, StyleGAN2, and bald StyleGAN2 which is StyleGAN2 without noise injection and path length regularizer. On DCGAN the proposed fuzzy reparameterization (FR) outperforms DCGAN with and without additive noise on FFHQ, CIFAR10, LSUN-Church measured with the Path Perceptual Length (PPL) and the FID. Also StyleGAN2 with FR outperforms StyleGAN2 with additive noise and bald StyleGAN2 on FFHQ and LSUN objects. To test numerical stability, condition numbers of 50000 Input, Pertubation pairs were computed for StyleGAN2 models with, without additive noise and with FR and path length regularization. StyleGAN2 + FR outperforms both on the mean and top-1000 mean condition number indicating that FR improves numerical stability. StyleGAN2 + FR also outperforms on the image inversion experiments.

Pros: This paper provides a theoretical framework for noise injection for GANs which is novel and interesting for the GAN community. The experimental results are extensive and convincing and support the theoretical analysis.

Cons: In section 4.3 the algorithm eq. 6-8 is not very clear to me. E.g. what are the parameters A,b, alpha and r and how are they motivated? PixSum is over the feature maps? A more detailed description with comments would be helpful for the reader. I could not find the FR implementation in the supplementary file, it looks like it contains only the original StyleGAN(2), DCGAN and DCGAN with additive noise models.

---

> ### Author Response · Authors · 2020-11-18
> **To Reviewer #2**
>
> To Reviewer #2:
> The major motivation behind eq. 6-8 is inspired by the deviation map in Fig. 1. The variance (sigma) of proposed noise injection method should model the local variance in the data manifold, which in its semantic meaning, is the detailed part of synthesis images, such as hair, parts of background, and silhouettes. The standard deviation map in Fig. 1 visualizes the deviation of sum of channels in the feature maps of StyleGAN blocks, which exactly corresponds to the semantic meaning of sigma. “A” is a learnable matrix which element-wisely controls the weight of contributions from each pixel of the feature map. “b”  is a learnable bias matrix to further adjust the bias of sigma. “r” is a scalar which controls the volume of injected noise as suggested in Theorems 2&3&4 to maintain locally Lipschitz property. Alpha is also a learnable scalar to explicitly regularize the structure of noise injection distribution and make sure that sigma is not ill-conditioned by adding an identity matrix to it., The matrices A and b in fact serve as a spatial attention enhancer to adjust the semantic attention in sigma. The proposed noise injection method is applied in each block of the StyleGAN2 network, thus resulting in a hierarchical adjustment to the detail information in the final outputs.

---

### Official Review · AnonReviewer3 · 2020-10-29
**Nice Results**

**Rating:** 7
**Confidence:** 2

**Review:**

To summarize, this paper proposed a new noise injection method that is easy to implement and is able to replace the original noise injection method in StyleGAN 2. The approach is supported by detailed theoretical analysis and impactful performance improvement on GAN training and inversion. The results show that they are able to achieve a considerable improvement on DCGAN and StyleGAN2.

Meanwhile, this reviewer did not fully understand the theoretical part and also has some questions regarding the implementation and results.

Based on my understanding, the fuzzy reparameterization technique realizes something that StyleGAN2 cannot achieve, and resolves some fundamental limitations of StyleGAN2. However, the improvement is not contiguous in Table 1, where we see the vanilla StyleGAN2 still outperforms the proposed architecture. What could be the reason?

FR seems to bring more parameters (Eq 6, 7, 8, 9). How many more compared to the additive noise implementation? Could the number of parameters be the reason that the proposed method performs better? Since FR can be seen as a generalization of StyleGAN2 noise injection, we would naturally expect that the proposed method should perform better than StyleGAN2. However, this is not always the case in Table 1. I guess more ablation studies can also be done on $\sigma$, such as interpolating between StyleGAN2 implementation and FR implementation, or a linear layer with the same number of additional parameters but has no constraint as in Eq. 6, 7, 8, 9.

For Figure 8, do we have the reconstruction visualization? Is the inversion done in the z space, w space or the w+ space?  I am curious to see how better this method performs in terms of inverting real images in the wild. I also believe the inversion in z space allows me to appreciate more about the inversion improvement.

Overall, I vote to accept this paper due to its good performance improvement over prior standard noise injection implementation. Meanwhile, I hope the theoretical analysis can be made easier to understand for a researcher that lacks the related background.

[Update after reading authors' comments]
Based on the authors' and other reviewers' comments,  I keep the score unchanged.

---

> ### Author Response · Authors · 2020-11-18
> **To Reviewer #3**
>
> To Reviewer #3:
>
> Concern1: Why vanilla StyleGAN2 outperforms the proposed architecture in FID score?
>
> Visual quality of StyleGAN2 has already been very good. Generated images are so close to real ones that FID distance is not that meaningful for measuring synthesis quality. Instead, semantic consistency turns to be a dominant factor. This phenomenon is clearly revealed and analyzed by the authors of StyleGAN2 in Figures 4&13&14 and section 3 of their paper. PPL is capable of capturing such semantic consistency and the smoothness of the network, which is essential to network generalization. Therefore, PPL is more sensible for visual quality of high-fidelity images. Our proposed method promotes PPL by a large margin. In the FFHQ case, our method, even without the help of PPL regularizer, can obtain better PPL score than the StyleGAN2.
>
> In the DCGAN cases, the synthesis quality is low, such that the FID still dominates the measurement of image quality. We then find that our method promotes both PPL and FID by large margins.
>
> The reviewer may also consult Fig. 3 in our paper. The hair quality of synthesis images of vanilla StyleGAN2 is significantly lower than those enhanced models after it in Fig. 3. While it could still maintain the lowest FID score, we do not think this really justifies the synthesis quality.
>
> Concern2: How many more parameters compared to the additive noise?
>
> The StyleGAN2 generator (with additive noise injection) has 24525213 parameters, while our proposed method has 24612552 parameters. Our method yields 87339 more parameters than StyleGAN2, which is about 0.3% of the total volume of StyleGAN2. Such minor increasement cannot be the main reason for those significant improvements in PPL scores and other metrics.
>
> Concern3: Ablation study of sigma.
>
> We in fact conduct many experiments to verify the detailed form of FR. As the priority of this work is a theoretical analysis to the general GAN models, and a theoretical framework for the new design of noise injection, we did not include that part in the submitted paper. We will add those contents in the revised version in the coming days.
>
> Concern4: Motivation of the implementation of FR
>
> The major motivation behind eq. 6-8 is inspired by the deviation map in Fig. 1. The variance (sigma) of proposed noise injection method should model the local variance in the data manifold, which in its semantic meaning, is the detailed part of synthesis images, such as hair, parts of background, and silhouettes. The standard deviation map in Fig. 1 visualizes the deviation of sum of channels in the feature maps of StyleGAN blocks, which exactly corresponds to the semantic meaning of sigma. “A” is a learnable matrix which element-wisely controls the weight of contributions from each pixel of the feature map. “b”  is a learnable bias matrix to further adjust the bias of sigma. “r” is a scalar which controls the volume of injected noise as suggested in Theorems 2&3&4 to maintain locally Lipschitz property. Alpha is also a learnable scalar to explicitly regularize the structure of noise injection distribution and make sure that sigma is not ill-conditioned by adding an identity matrix to it., The matrices A and b in fact serve as a spatial attention enhancer to adjust the semantic attention in sigma. The proposed noise injection method is applied in each block of the StyleGAN2 network, thus resulting in a hierarchical adjustment to the detail information in the final outputs.
>
>
> Concern5: Where is inversion done?
>
> While this paper is not for the principle of GAN inversion, we simply use the Image2StyleGAN method to perform the inversion in the w+ space.

---

### Author Response · Authors · 2020-11-22
**To all**

To all:

We thank all the reviewers for their helpful comments. We have uploaded a revised version of our paper according to the suggestions of the reviewers. The revision mainly includes the following four aspects:


1. We rewrite the proposed FR method in section 4.3 with more detailed descriptions and motivations.


2. We add an ablation study in the appendix to support the motivations of FR implementation in section 4.3, which demonstrates the effectiveness of those procedures.


3. We add related citations to the PageRank algorithm we used in LSUN-Cat dataset in the appendix.


4. We add more explanation to the meaning and influence of Theorem 1 (adversarial dimension trap) to avoid possible misunderstandings.


We welcome any further questions or suggestions about our paper.

---

### Author Response · Authors · 2020-11-23
**To all**

We just updated the new version of our submission again. We further polished our writing in this version to make our paper more readable. We thank all the reviewers to help improve our paper.

---

### Decision · Program_Chairs · 2021-01-07
**Final Decision**

**Decision:**

Reject

**Comment:**

This paper studies the role of “noise injection” in GANs with tools from Riemannian geometry, and derives a new noise injection approach that aims to learn a fuzzy coordinate system to model non-Euclidean geometry. The new noise injection approach is shown to improve over StyleGANv2 noise injection on lower-resolution 128x128 FFHQ, LSUN, and 32x32 CIFAR-10 images.


Some reviewers found the experimental results a “considerable improvement on DCGAN and StyleGANv2” (R3), “extensive and convincing” (R2), while others had concerns around the experimental setup using lower resolution images (R1, R4).  While reviewers were mostly positive about the experimental wins of the paper, there was confusion (R3) and several concerns (R4) around the theory and the relationship between the theory and the practical noise injection algorithm. I additionally had several concerns around the presentation and relation to prior work on generative models. Thus in the current state I cannot recommend this paper for acceptance. Below I highlight concerns that should be addressed in future revisions.


1. My biggest concern is the tremendous gap between the theoretical claims and the practical implementation. When training a GAN with the new form of noise injection, does it learn the skeleton and fuzzy equivalence relationships you claim? This paper is missing any kind of toy experimenting showing that training a GAN with fuzzy reparameterization discovers these relationships or coordinates. Such an experiment would greatly strengthen the paper and help to answer the question of why this new method works (i.e. it’s not just more parameters, a slightly better architecture, or better hyperprameters as mentioned by R3 and R4). There’s also no discussion of what happens theoretically when you have multiple layers of fuzzy reparameterization, and the claims that StyleGAN2’s noise injection limits to Euclidean geometry is false in this case (and thus StyleGAN2’s noise injection can also overcome the “adversarial dimension trap”).

2. Theoretical setting: As mentioned by R4, there is much prior work on the difficulties in fitting a lower-dimensional model manifold to a higher-dimensional data manifold (e.g. WGAN). Theorem 1 highlights the impossibility of exactly fitting the data manifold with (smooth) neural networks, but the resulting solutions of increasing the dimensionality of the latent space is well-known and commonly used (e.g. StyleGAN). This paper also doesn’t discuss the alternative of *approximately* fitting the data manifold with a lower-dimensional structure, which is what is often studied in practice.


3. Clarity: The term “noise injection” is overloaded in the literature, and the current presentation of the paper does not sufficiently describe the method. There’s also no discussion of “instance noise” that is another solution to this problem that adds noise to inputs of the discriminator to yield finite f-divergences (Sonderby et al., 2016, Roth et al., 2017). The work on instance noise is very related to the approach here, but only adds noise to the output of the generator, not at all levels.
There's also no discussion of how adding noise is just expanding the generative model with additional latent variables, a standard approach that is often discussed in the context of hierarchical generative models. The authors mention the relation to reparameterization trick in VAEs, but argue it is doing something fundamentally different. However, modern VAE architectures (IAF-VAE, Very Deep VAE), use a very similar form of modulation at multiple levels in the hierarchy.


4. Experiments: There are no error bars in experimental results, and many results are presented in a new experimental setting defined by the authors (lower resolution than prior work even if using prior code). Rerunning experiments in more standard settings on full resolution images would greatly improve the confidence that the new noise injection strategy is effective.